# Structurally Aligned Subtask-Level Memory for Software Engineering Agents

**Kangning Shen** [1]  **Jingyuan Zhang** [1]  **Chenxi Sun** [1]  **Wencong Zeng** [1]  **Yang Yue** [1]

## Abstract

Large Language Models (LLMs) have demonstrated significant potential as autonomous software engineering (SWE) agents. Recent work has further explored augmenting these agents with memory mechanisms to support long-horizon reasoning. However, these approaches typically operate at a coarse instance granularity, treating the entire problem-solving episode as the atomic unit of storage and retrieval. We empirically demonstrate that instance-level memory suffers from a fundamental granularity mismatch, resulting in misguided retrieval when tasks with similar surface descriptions require distinct reasoning logic at specific stages. To address this, we propose Structurally Aligned Subtask-Level Memory, a method that aligns memory storage, retrieval, and updating with the agent's functional decomposition. Extensive experiments on SWE-bench Verified demonstrate that our method consistently outperforms both vanilla agents and strong instance-level memory baselines across diverse backbones, improving mean Pass@1 over the vanilla agent by +4.7 pp on average (e.g., +6.8 pp on Gemini 2.5 Pro). Performance gains grow with more interaction steps, showing that leveraging past experience benefits long-horizon reasoning in complex software engineering tasks.

## 1. Introduction

Large Language Models (LLMs) have demonstrated significant potential as autonomous agents for software engineering (SWE), particularly in resolving repository-level issues through multi-turn interactions with execution environments (Qwen Team, 2025; Zeng et al., 2025; Chen et al., 2025a). Unlike simple code completion, software engineering tasks are distinctive in that successful agents typically follow a

[1]Kuaishou Technology, Beijing, China. Correspondence to: Yang Yue <yyangyue01@gmail.com>.

*Proceedings of the 43rd International Conference on Machine Learning*, Seoul, South Korea. PMLR 306, 2026. Copyright 2026 by the author(s).

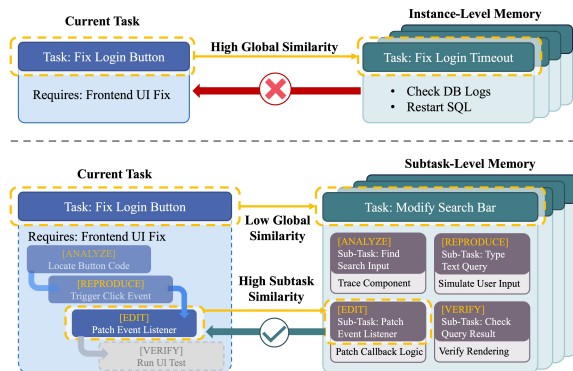

*Figure 1.* **Comparison of Instance-Level vs. Subtask-Level Memory. Top:** Instance-level memory relies on global task similarity, causing reasoning interference when tasks share surface goals but require different reasoning logic. **Bottom:** Our method retrieves by matching stage-consistent subtask intents, enabling precise experience reuse even across globally dissimilar tasks.

structured workflow composed of heterogeneous subtasks, such as analyzing the problem description, reproducing the bug, localizing the fault, editing code, and validating the fix (Yang et al., 2024a; Xia et al., 2025; Applis et al., 2025).

Motivated by these characteristics, recent work (Park et al., 2023; Shinn et al., 2023) has begun to explore memory-aware agent systems that enable learning from and reuse of past experiences in software engineering tasks. However, these approaches typically operate at a coarse instance granularity, treating an entire problem-solving episode as the atomic unit of storage and retrieval (Ouyang et al., 2025; Mu et al., 2025). Despite their success, such designs rely heavily on global task similarity, implicitly assuming that similarity at the level of overall issue descriptions is sufficient to ensure the transferability of historical solution trajectories in their entirety (Xiong et al., 2025; Chen et al., 2025b).

However, this monolithic treatment is fundamentally misaligned with the compositional nature of software engineering (Xia et al., 2024), creating a granularity mismatch where global similarity fails to capture stage-specific reasoning needs (Lin et al., 2025). As shown in Figure 1, global similarity is neither sufficient nor necessary for effective local reasoning. Issues such as "Fix Login Button" and "Fix Login Timeout" appear highly similar on the surface, as both concern the login functionality. However, they require entirely different reasoning paths: the former often involves

frontend UI logic, while the latter typically requires debugging backend services or database interactions. Retrieving experience based solely on global similarity can therefore introduce irrelevant guidance and mislead the agent (Du et al., 2025). Conversely, globally dissimilar tasks such as "Modify Search Bar" and "Fix Login Button" may share the same subtask operation (e.g., updating a frontend event listener). Instance-level memory mechanisms fail to capture such reusable skills because low global similarity prevents retrieval. To resolve both failure modes, memory retrieval should align with the functional granularity of the reasoning process rather than the global problem definition (Zhou et al., 2024; Zhao et al., 2025).

To bridge this gap, we propose Structurally Aligned Subtask-Level Memory, a method that aligns memory operations with the agent's functional decomposition. As shown in the bottom of Figure 1, we explicitly model the reasoning process as a sequence of discrete subtasks, defined as atomic units of reasoning labeled by functional categories (e.g., ANALYZE, EDIT). Unlike instance-level storage, our memory state acts as a repository of fine-grained subtask entries, where each entry is structured as a triple $(z, d, e)$ comprising the category $z$, a localized intent description $d$, and the abstracted experience $e$. During retrieval, our method performs a two-stage retrieval strategy: it first enforces the category $z$ as a hard constraint to filter cross-stage noise, and subsequently retrieves the specific experience that semantically matches the current intent $d$. Upon subtask completion, the method abstracts transferable insights from the raw subtask trajectory and incrementally updates the memory state, ensuring that experience accumulation remains aligned with the reasoning granularity.

We evaluate the proposed method on SWE-bench Verified (Jimenez et al., 2024) across four diverse backbone models (Comanici et al., 2025; Anthropic, 2025a;b), comparing against vanilla agents (Yang et al., 2024a) and strong instance-level memory baselines (Ouyang et al., 2025). Experimental results demonstrate consistent gains over both vanilla agents and instance-level memory baselines, with absolute Pass@1 improvements over the vanilla agent of up to +6.8 pp on Gemini 2.5 Pro and +4.7 pp on average across all evaluated backbones. Further analysis reveals that the performance gain becomes more pronounced as the number of interaction steps increases, indicating that incorporating past experience is particularly beneficial for enabling more effective long-horizon reasoning in complex software engineering tasks.

## 2. Related Work

**LLM Agents for Software Engineering.** Code generation has evolved from isolated functions (Chen, 2021; Austin et al., 2021; Hendrycks et al., 2021) to complex repository-level challenges (Liu et al., 2023; Li et al., 2024; 2025). For example, Jimenez et al. (2024) introduced SWE-bench to systematically assess the ability of LLMs to resolve real-world GitHub issues, exposing a significant gap between current models and practical engineering requirements, with extensions that incorporate multimodal evidence (Yang et al., 2024b), continual-learning evaluation (Joshi et al., 2025), and multilingual issue-resolving settings (Zan et al., 2025). Motivated by these challenges, SWE-agent (Yang et al., 2024a) introduces an Agent-Computer Interface for repository interaction. Recent SWE agents span hierarchical long-horizon planning and interactive platforms with tool-augmented execution (Xu et al., 2025b; Wang et al., 2024; Zhang et al., 2024). Complementary lines show that static workflows and search-based pipelines can also be competitive (Xia et al., 2024; Antoniades et al., 2024). Furthermore, code-specialized model releases and technical reports leverage code-centric post-training to improve downstream tool use (Liu et al., 2024a; Cognition AI, 2025). Despite these advances, existing agents still suffer from non-uniform long-context utilization and short-term memory bottlenecks when facing long-range reasoning and cross-file dependencies (Liu et al., 2024b). Consequently, structure-aligned, retrievable memory is crucial for robust and efficient software engineering agents.

**Memory Mechanisms for Agents.** To capture effective strategies from past interactions, recent work increasingly formalizes agent memory as structured components and scalable infrastructure (Sumers et al., 2023; Hu et al., 2025). In software engineering settings, a representative line distills prior agent trajectories into reusable experience or lessons for test-time reuse (Ouyang et al., 2025; Chen et al., 2025b; Mu et al., 2025). A complementary direction constructs repository-anchored memory from project artifacts (e.g., commit histories) to provide persistent repository knowledge for downstream reasoning, which is particularly useful for code localization and long-horizon debugging in large codebases (Wang et al., 2025; Cubranic et al., 2005; Singh et al., 2025). Other work targets memory governance and scalability, spanning curated experience cards, gated cross-agent sharing, and distributed memory for long-running (often multi-agent) systems (Wang et al., 2026; Tang et al., 2025; Xu et al., 2025a; Yuen et al., 2025). While effective, most prior designs operate at the episode/instance or repository/global granularity and rely on global similarity as the primary transfer signal, which can introduce cross-stage noise or miss reusable stage-specific skills. Our work differs by (i) shifting the memory unit to fine-grained experience, and (ii) updating memory online by abstracting actionable experiences from completed subtasks, directly addressing the granularity mismatch between episodic memory and stage-structured reasoning in SWE agents.

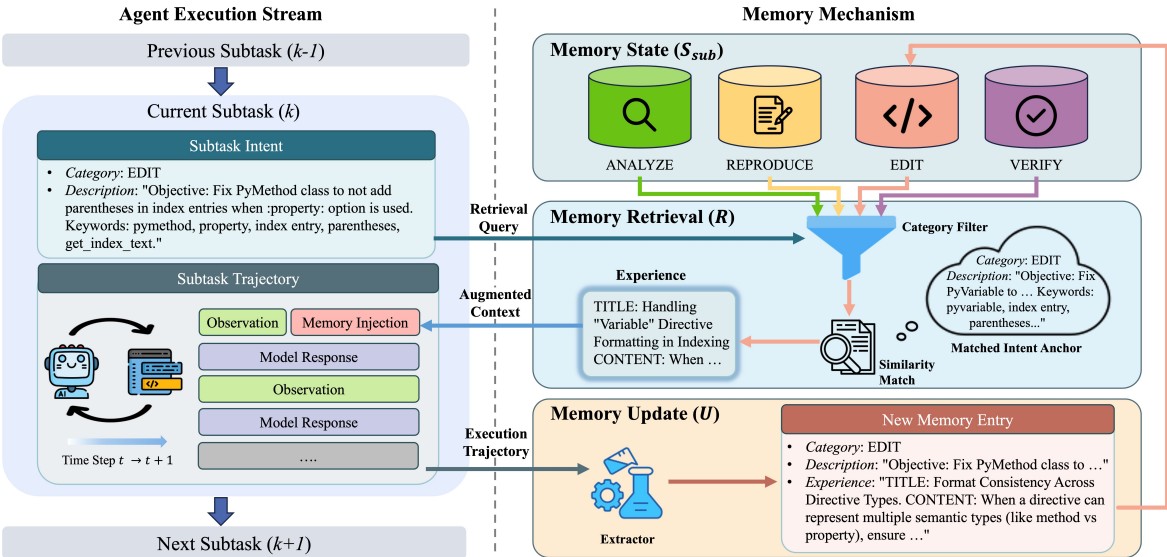

*Figure 2.* **Overview of the Structurally Aligned Subtask-Level Memory method.** Unlike instance-level approaches, our method aligns memory operations with the agent's functional decomposition (e.g., EDIT). The process operates in two key phases: (1) **Retrieval ($R$):** Initialized by the *Subtask Intent*, the agent queries the Memory State ($S_{\text{sub}}$). A *Category Filter* followed by a *Similarity Match* retrieves a structurally relevant historical anchor to provide *Augmented Context*. (2) **Update ($U$):** The *Subtask Trajectory* is processed by an *Extractor* to summarize abstract, transferable insights, which are stored as a *New Memory Entry*.

## 3. Method

### 3.1. Task Formulation and Structural Alignment

We define a software engineering task as $T = (\mathcal{C}, \mathcal{G})$, comprising a codebase $\mathcal{C}$ and a goal $\mathcal{G}$ ([Jimenez et al., 2024](#)). An agent interacts with the environment to produce a trajectory $\pi = (s_1, a_1, \ldots, s_T)$. While typically modeled as a flat sequence, real-world issue resolution inherently entails a compositional structure. We therefore posit that a trajectory $\pi$ admits a latent decomposition into $K$ reasoning-aligned subtasks, denoted as $\pi = \pi^{(1)} \oplus \cdots \oplus \pi^{(K)}$, where each segment $\pi^{(k)}$ corresponds to a discrete functional unit.

To support long-horizon reasoning, we augment the agent with a memory mechanism, formally defined as a tuple $\mathcal{M} = (S, R, U)$ ([Packer et al., 2023](#)). Here, $S$ denotes the memory state accumulating past experiences; $R(h_t, S) \to c_t$ is the retrieval function that maps the current history $h_t$ to relevant context $c_t$; and $U(S, \pi_{new}) \to S'$ is the update function that evolves the state with new trajectories. Recent instance-level approaches ([Ouyang et al., 2025](#); [Mu et al., 2025](#)) typically instantiate $\mathcal{M}$ by treating the global trajectory $\pi$ as the atomic unit of storage ($S_{inst} = \{(\tau, E_\tau)\}$). We identify a fundamental granularity mismatch in this formulation: software issues often exhibit a *one-to-many mapping* from surface symptoms to root causes. As illustrated in Figure 1, relying on global task similarity fails to capture stage-specific reasoning requirements, leading to reasoning interference when surface-level resemblances mask distinct underlying logic.

To mitigate this mismatch, we propose a structurally aligned subtask-level memory method in which memory is retrieved and updated at the same functional granularity as the agent's reasoning, i.e., on decomposed subtask units $\pi^{(k)}$ rather than monolithic instance-level trajectories. Figure 2 provides an end-to-end overview of the memory-augmented agent workflow. Section 3.2 defines the subtask formulation, while Sections 3.3–3.5 describe the memory components. Algorithm 1 summarizes the overall workflow.

### 3.2. Subtask Modeling via Dynamic Segmentation

To operationalize structural alignment, we partition the reasoning process into a set of disjoint functional categories, defined as the set $Z = \{\text{ANALYZE}, \text{REPRODUCE}, \text{EDIT}, \text{VERIFY}\}$. Each category $z \in Z$ corresponds to a distinct mode of software engineering reasoning, reflecting the heterogeneous functional requirements at different stages of the resolution process. This category set is a task-specific instantiation for repository-level bug fixing rather than a universal workflow ontology. More generally, our framework only requires a functional category space that separates reasoning modes with different memory needs, while keeping the same subtask representation, category-constrained retrieval, and experience-update mechanism. For other software engineering tasks, the category space can therefore be adapted to the task structure, as we later examine in the feature-implementation setting.

**Algorithm 1** Subtask-Level Memory Mechanism

---

1: **Input:** Task $\tau$, Memory State $S_{\text{sub}}$, LLM $\pi_\theta$, Extractor $\mathcal{E}$
2: Initialize global trajectory $H \leftarrow \emptyset$, subtask index $k \leftarrow 1$
3: Predict initial intent $(z^{(1)}, d^{(1)})$ based on task goals
4: **while** Task $\tau$ not completed **do**
5:    // Phase 1: Contextual Retrieval
6:    Retrieve $m^* \leftarrow \text{Top-1}\left(d^{(k)}, \{m \in S_{\text{sub}} \mid m.z = z^{(k)}\}\right)$
7:    **Augment context:** $c_{\text{aug}}^{(k)} \leftarrow [\, m^*.e\,;\,\tau\,]$
8:    // Phase 2: Execution Loop
9:    Initialize local trajectory $\pi^{(k)} \leftarrow \emptyset$
10:    **repeat**
11:      Sample action $a_t \sim \pi_\theta(\cdot \mid c_{\text{aug}}^{(k)}, H)$
12:      Execute $a_t$, observe $s_{t+1}$
13:      $H \leftarrow H \oplus (a_t, s_{t+1})$;    $\pi^{(k)} \leftarrow \pi^{(k)} \oplus (a_t, s_{t+1})$
14:    **until** Agent signals completion of subtask $z^{(k)}$
15:    // Phase 3: Reflection-Based Update
16:    Extract experience: $e^{(k)} \leftarrow \mathcal{E}(\pi^{(k)}, z^{(k)}, d^{(k)})$
17:    Store entry: $S_{\text{sub}} \leftarrow S_{\text{sub}} \cup \{(z^{(k)}, d^{(k)}, e^{(k)})\}$
18:    // Transition
19:    Predict next intent $(z^{(k+1)}, d^{(k+1)})$ and increment $k$
20: **end while**

---

We formalize the fundamental unit of reasoning as a subtask tuple $\phi^{(k)} = (z^{(k)}, d^{(k)}, \pi^{(k)})$. Crucially, the pair $(z^{(k)}, d^{(k)})$ constitutes the subtask intent, which encapsulates the agent's planning state. Here, $z^{(k)}$ is the functional category, while $d^{(k)}$ serves as a structured description explicitly formatted to include two components: an objective (synthesized from the current context to define the immediate sub-goal) and keywords (capturing mechanism-level entities or evidence). This structured format decouples local reasoning requirements from the global task description. $\pi^{(k)}$ denotes the execution trajectory segment associated with this intent.

To instantiate these subtasks within a continuous interaction stream, we employ a transition-oriented segmentation strategy. Integrating transition logic directly into the system prompt allows the agent to autonomously monitor its progress. Upon concluding a reasoning phase (subtask $k$), the agent explicitly predicts the next category $z^{(k+1)}$ and synthesizes the corresponding description $d^{(k+1)}$ as part of its thought process. This design introduces minimal architectural overhead, as the segmentation is woven directly into the agent's standard reasoning stream via autonomous prediction.

### 3.3. Structured Memory State

The state component $S_{\text{sub}}$ functions as a repository of accumulated subtask experiences, explicitly organizing them as Problem-Solution mappings. Each memory entry $m \in S_{\text{sub}}$ is formalized as a structured triple:

$$m = (z, d, e). \qquad (1)$$

- **Category ($z$):** The functional label (e.g., ANALYZE)

indicating the specific reasoning phase to which this experience belongs.

- **Description ($d$):** A structured abstraction of the problem context prior to resolution, explicitly capturing the subtask's objective and keywords.

- **Experience ($e$):** The actionable insights abstracted from the subtask trajectory segment $\pi^{(k)}$. Unlike raw logs which contain instance-specific noise, $e$ retains only the critical steps, guidelines, or behavioral patterns necessary to resolve similar subtasks.

### 3.4. Contextual Retrieval

Retrieval is triggered at the initialization of each subtask $\phi^{(k)}$ to condition the agent's reasoning on relevant historical experiences. This process follows a hierarchical strategy of category-based filtering followed by semantic matching.

**Category-Based Filtering.** The mechanism restricts the retrieval search space strictly to entries matching the current category $z^{(k)}$. This explicit partitioning drastically reduces the candidate pool, enforcing structural alignment, ensuring that the agent only accesses experiences within the same functional scope (e.g., excluding debugging heuristics during a code-editing phase).

**Semantic Matching.** Within the filtered candidate set, the mechanism retrieves the most relevant memory entry by measuring the semantic similarity between the current subtask description $d^{(k)}$ and the stored anchors $m.d$. Formally, the optimal memory entry $m^*$ is identified as:

$$m^* = \underset{m \in S_{\text{sub}},\, m.z = z^{(k)}}{\arg\max} \cos\big(E(d^{(k)}), E(m.d)\big), \quad (2)$$

where $E(\cdot)$ denotes a fixed embedding model.

**Memory Injection.** The retrieved experience content $m^*.e$ is incorporated into the subtask's initial context, forming an augmented observation $\tilde{s}_0^{(k)}$. Conditioned on this augmented context, the agent can transfer knowledge at test time without any parameter updates.

### 3.5. Experience Accumulation

To enable continuous self-evolution, the system executes a structured update upon the conclusion of each subtask $\phi^{(k)}$, allowing the agent to continuously accumulate reusable experiences online.

**Experience Extraction.** We employ an LLM-based operator $\mathcal{E}$ to distill the subtask trajectory $\pi^{(k)}$ into transferable experience $e^{(k)}$ via a two-stage process. First, $\mathcal{E}$ evaluates the correctness of the subtask execution. This judgment

*Table 1.* Main results on SWE-bench Verified. We report the Best@3 (the maximum Pass@1 score across three independent runs) and the Avg. Pass@1 (mean $\pm$ sample std. dev.). Inst-Mem refers to the instance-level memory baseline reproduced from Ouyang et al. (2025). $\Delta$ and Rel. denote the absolute and relative improvement of our subtask-level memory method over the Vanilla baseline.

| Backbone Model | Vanilla Agent | | Instance-level Mem | | Subtask-level Mem (Ours) | | Improvement ($\Delta$) | |
|---|---|---|---|---|---|---|---|---|
| | Best@3 | Avg. Pass@1 | Best@3 | Avg. Pass@1 | Best@3 | Avg. Pass@1 | Abs. | Rel. |
| Gemini 2.5 Flash | 36.8 | $36.3 \pm 0.76$ | 39.8 | $37.8 \pm 2.00$ | **43.2** | $\mathbf{41.9} \pm \mathbf{1.17}$ | +5.6 | +15.4% |
| Gemini 2.5 Pro | 54.8 | $53.5 \pm 1.21$ | 56.0 | $55.1 \pm 1.50$ | **61.2** | $\mathbf{60.3} \pm \mathbf{1.29}$ | +6.8 | +12.7% |
| Claude 3.7 Sonnet | 52.8 | $52.2 \pm 0.72$ | 53.6 | $51.1 \pm 2.27$ | **57.2** | $\mathbf{56.1} \pm \mathbf{1.33}$ | +3.9 | +7.5% |
| Claude 4.0 Sonnet | 64.4 | $63.5 \pm 0.81$ | 63.8 | $63.3 \pm 0.81$ | **66.8** | $\mathbf{65.8} \pm \mathbf{0.92}$ | +2.3 | +3.6% |

guides the subsequent extraction, allowing $\mathcal{E}$ to distinguish between different types of insights: it identifies successful patterns from correct trajectories while distilling failure avoidance strategies from erroneous ones. Throughout this process, the operator filters out repository-specific noise (e.g., file paths) to retain only actionable insights. Crucially, $\mathcal{E}$ uses the same LLM backbone as the task-solving agent, ensuring that performance gains arise from the memory mechanism rather than a stronger teacher model.

**Memory Incrementation.** The extracted experience $e$ is paired with the subtask intent to form a new memory entry $m_{\text{new}}$, structured as the triple defined in Section 3.3:

$$m_{\text{new}} = (z^{(k)}, d^{(k)}, e^{(k)}). \tag{3}$$

This new entry is appended to the state component: $S_{\text{sub}} \leftarrow S_{\text{sub}} \cup \{m_{\text{new}}\}$. This cycle closes the loop, allowing the system to incrementally refine its knowledge base and replicate successful reasoning patterns in subsequent tasks.

# 4. Experiments

## 4.1. Experimental Setup

**Benchmark, Metrics, and Agent Scaffold.** We evaluate our method on SWE-bench Verified (Jimenez et al., 2024; OpenAI, 2024), a rigorous benchmark comprising 500 real-world GitHub issues. Performance is reported using Pass@1, measuring the percentage of instances where the agent generates a correct patch in a single attempt. To strictly isolate the impact of memory mechanisms, all methods are implemented on the Mini SWE Agent scaffold (Yang et al., 2024a; SWE-bench Team, 2025) utilizing the official system prompts. We employ greedy decoding (`temp = 0`) for the agent policy to minimize stochasticity, while maintaining identical non-memory configurations (e.g., execution environment, step limits) across all experiments. Crucially, for our method, the per-instance step limit encompasses both the agent's reasoning steps and the memory extraction overhead, ensuring a budget-neutral comparison.

**Baselines, Models, and Protocol.** We compare our approach against two primary baselines: (i) Vanilla Agent: the

standard Mini SWE Agent without memory; (ii) Instance-level Memory: a faithful reproduction of Reasoning-Bank (Ouyang et al., 2025), which stores and retrieves instance-level reasoning summaries based on global semantic similarity, and updates the memory after each instance.

To demonstrate robustness across varying model capabilities, we evaluate four backbone LLMs: Gemini 2.5 Flash, Gemini 2.5 Pro (Comanici et al., 2025), Claude 3.7 Sonnet (Anthropic, 2025a), and Claude 4.0 Sonnet (Anthropic, 2025b). All experiments follow a test-time streaming protocol: the memory storage $S_{\text{sub}}$ is initialized as empty and accumulates experience on-the-fly. To mitigate ordering effects inherent to streaming, we perform three independent runs, using distinct random seeds to shuffle the execution sequence of the 500 instances, and report Avg. Pass@1 as mean $\pm$ std over three runs; we additionally report Best@3 (the best run among the three) as an upper bound.

## 4.2. Main Results

We present the comparative performance of our Subtask-level Memory method against the Vanilla and Instance-level baselines across four distinct backbone models on SWE-bench Verified.

**Effectiveness and Robustness.** As detailed in Table 1, our Subtask-level Memory achieves consistent gains across all evaluated backbones, outperforming both vanilla agents and standard retrieval baselines. Our method demonstrates robust generalizability across model scales, yielding consistent improvements on both lightweight models (Gemini 2.5 Flash, +5.6 pp) and frontier reasoning models (Claude 4.0 Sonnet, +2.3 pp). Across all four backbones, our method improves Pass@1 by +4.7 pp on average. Crucially, the improvements are consistent across all backbones and seeds, indicating that fine-grained experience accumulation generalizes beyond a particular model family. We further observe reduced run-to-run variability compared to instance-level retrieval, suggesting improved robustness under the streaming protocol.

**Mitigating Reasoning Interference.** A critical observation lies in the instability of the Instance-level Memory baseline. While effective for Gemini models, it suffers from severe robustness issues on the Claude family: performance degrades on Claude 3.7 Sonnet ($52.2\% \rightarrow 51.1\%$) and stagnates on Claude 4.0 Sonnet ($63.5\% \rightarrow 63.3\%$). The high variance observed on Claude 3.7 Sonnet ($\pm 2.27$) further indicates that coarse-grained memory introduces irrelevant, instance-specific noise that confuses capable reasoners. In sharp contrast, our approach outperforms the Instance-level baseline across all models, effectively stabilizing the high variance observed in three backbones (e.g., reducing $\sigma$ from 2.27 to 1.33 on Claude 3.7 Sonnet). By isolating experiences into functional subtasks, our method filters out contextual noise, ensuring that stronger models leverage retrieved knowledge effectively without being misled by spurious correlations.

### 4.3. Ablation Studies

In this section, we present a series of ablation studies to dissect the individual contributions of each core component in our method. Specifically, we systematically investigate: (1) the source of performance gains by decoupling the structured workflow from the memory content; (2) the necessity of category isolation in preventing retrieval interference; and (3) the impact of experience abstraction versus raw interaction history. All experiments are conducted using Claude 3.7 Sonnet on the full task stream of SWE-bench Verified.

**Impact of Structured Decomposition.** To determine whether performance gains stem from accumulated experience or merely the structured workflow (i.e., the forced multi-stage decomposition), we evaluate a Structure-only control. In this setting, the agent is explicitly prompted to adhere to the structured workflow and autonomously determine subtask transitions, yet operates without any memory retrieval or update mechanisms.

*Table 2.* Ablation: With Memory vs. Structured Prompting Only.

| Configuration | Pass@1 (%) | $\Delta$ |
|---|---|---|
| Vanilla Agent | 52.2 | - |
| Structured prompting only | 53.2 | +1.0 |
| **Ours (Full Method)** | **56.1** | **+3.9** |

As shown in Table 2, structural scaffolding alone yields only a modest improvement (+1.0%) over the Vanilla baseline, suggesting that modern LLMs already possess strong implicit planning capabilities. In contrast, the full method delivers a substantial performance leap (+3.9%). This confirms that the structural constraint primarily serves as an alignment index for retrieval, while the decisive factor is the transferable experience injected from the memory state.

**Impact of Category Isolation.** To validate the necessity of isolating memories by subtask categories, we compare our category isolated method against a variant without category filtering. In this baseline, the retrieval mechanism performs a global search across the entire collection of accumulated memories to select the top-1 experience based on similarity, bypassing the category-specific constraint utilized in our full method.

*Table 3.* Ablation: Category Isolation vs. Global Retrieval.

| Retrieval Strategy | Pass@1 (%) | $\Delta$ |
|---|---|---|
| Vanilla Agent | 52.2 | - |
| No Category Filter | 53.8 | +1.6 |
| **Ours (Category Isolated)** | **56.1** | **+3.9** |

As shown in Table 3, while global retrieval yields a modest improvement (+1.6%) over the Vanilla baseline, it significantly underperforms the category-isolated model (+3.9%). This performance gap highlights that structural precision is critical. Given the Top-1 retrieval constraint, selecting a semantically similar but functionally irrelevant memory from a global pool introduces fatal noise. By enforcing category isolation, our method ensures that the retrieved insight is functionally aligned with the agent's current objective.

**Necessity of Abstraction.** To assess the utility of the extraction operator $\mathcal{E}$, we compare our method against a Raw Trajectory variant that stores and injects the whole interaction history directly as memory content.

*Table 4.* Ablation: Abstract Insights vs. Raw Trajectories.

| Memory Content | Pass@1 (%) | $\Delta$ |
|---|---|---|
| Vanilla Agent | 52.2 | - |
| Raw Trajectory | 53.4 | +1.2 |
| **Ours (Abstract Insight)** | **56.1** | **+3.9** |

As shown in Table 4, replacing insights with raw trajectories significantly degrades performance. The marginal gain of Raw Trajectory (+1.2%) suggests that instance-specific artifacts (e.g., file paths) in raw traces act as noise, hindering adaptation to new contexts. In contrast, the extraction operator (+3.9%) functions as a semantic filter, isolating generalizable strategies from execution details. This confirms that abstraction is strictly necessary to decouple reasoning logic from task-specific noise.

### 4.4. Analysis

In this section, we analyze the internal mechanisms driving the performance of our method, utilizing a representative execution of Claude 3.7 Sonnet. We specifically investigate the temporal dynamics of online learning, the system's robustness across varying task complexities, and the func-

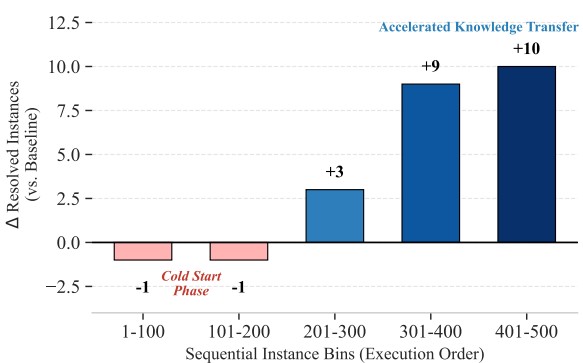

*Figure 3*. Temporal Dynamics of Experience Accumulation. Net gain (Δ Resolved) relative to the baseline across sequential bins of 100 instances. The trend illustrates the transition from a sparse memory state (1-200) to accelerated knowledge transfer (301-500) as experience accumulates.

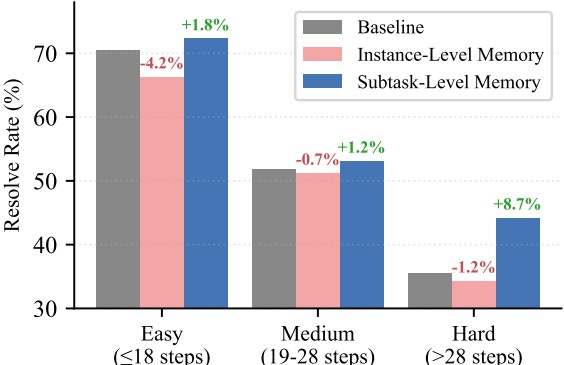

*Figure 4*. Pass@1 Improvement by Complexity. Instances are grouped by baseline trajectory length (step count). The method delivers disproportionate gains (+8.7%) on Hard tasks ($> 28$ steps), demonstrating its utility in mitigating long-horizon reasoning failures compared to the baseline.

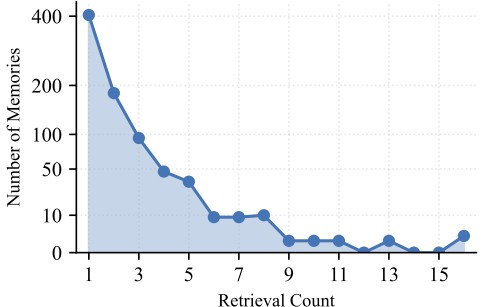

*Figure 5*. Memory Retrieval Frequency. The distribution follows a long-tail pattern: a small "head" of generic memories is retrieved frequently, while a vast "tail" of over 400 single-use memories captures instance-specific edge cases.

tional characteristics of the accumulated knowledge. These analyses provide insights into how the method adapts to the software engineering domain beyond aggregate metrics.

**Temporal Dynamics of Online Learning.** To investigate the efficacy of experience accumulation, we analyze the improvement curve of a representative run in Figure 3. Instead of aggregate metrics, we plot the net performance gain (Δ Resolved) relative to the Vanilla baseline across five sequential buckets of 100 instances.

The trend demonstrates a clear correlation between the density of the memory state $S_{sub}$ and agent performance. In the initial buckets (Instances 1-200), the memory state is sparse; the agent exhibits a slight performance dip ($-1$) due to the overhead of retrieval without sufficient relevant matches. As execution proceeds (201-300), we observe a moderate recovery ($+3$) as the system begins to capture actionable patterns. Crucially, as the memory becomes sufficiently populated in the final stages (301-500), the agent achieves substantial gains ($+9$ and $+10$). This continuous upward trend confirms that the method effectively leverages accumulated history to "shortcut" reasoning in later tasks, validating the hypothesis that performance scales with the richness of the memory state.

**Efficacy Across Task Complexity.** We stratify the test instances into Easy, Medium, and Hard tiers based on the baseline agent's trajectory length (step count), serving as a proxy for problem complexity. The split yields roughly balanced groups: Easy ($\leq 18$ steps, $n$=166), Medium (19-28 steps, $n$=162), and Hard ($> 28$ steps, $n$=172).

Figure 4 presents the Pass@1 comparison. The results reveal a complexity-dependent gain: while the improvement on Easy tasks is marginal (+1.8%), the method delivers a substantial boost on Hard instances, increasing the success rate by +8.7% (from 35.5% to 44.2%).

This distribution supports two critical conclusions: (1) High-Complexity Utility: Hard instances typically involve prolonged trial-and-error loops or obscure environment configurations. In these scenarios, subtask memories act as computational shortcuts, providing precise reproduction scripts or API usages that "short-circuit" costly exploration, effectively preventing the agent from getting lost in long trajectories. (2) Robustness on Simple Tasks: For Easy instances, the baseline model already possesses sufficient internal knowledge, leading to a ceiling effect. However, the performance does not regress, confirming that our category-routed retrieval is precise enough to avoid introducing distracting noise even when the task is trivial.

**Memory Distribution and Diversity.** We analyze the utilization patterns of the 797 unique memory entries retrieved across the 500 test instances. As shown in Figure 5, the usage frequency exhibits a long-tail distribution. The "head" (top 100 memories) accounts for 33.8% of retrievals, rep-

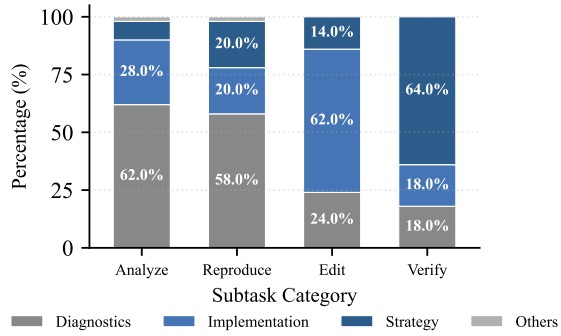

*Figure 6.* Functional Distribution of Memories. The semantic category of retrieved memories shifts dynamically across phases. *Diagnostics* dominate the ANALYZE phase (62.0%), shifting to *Implementation* during EDIT (62.0%) and *Strategy* during VERIFY (64.0%).

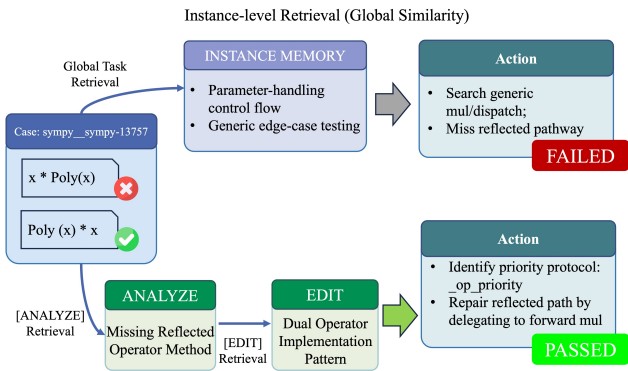

*Figure 7.* Case study on `sympy__sympy-13757`. While the instance-level baseline fails due to coarse-grained retrieval of generic concepts, our method retrieves precise, subtask-specific insights to successfully resolve the reflected multiplication issue.

resenting recurring general strategies (e.g., specific library quirks). Conversely, the "tail" is significant: 50.7% of memories are single-use. This confirms that the agent does not merely overfit to a few generic rules but maintains a vast library of instance-specific insights to handle corner cases.

**Functional Specialization of Knowledge.** To characterize the nature of the learned insights, we employ Gemini-2.5-Pro (Comanici et al., 2025) to categorize 200 memory entries sampled across four phases. As illustrated in Figure 6, the distribution reveals a distinct functional specialization that aligns with the task-solving lifecycle: (1) Problem Diagnosis: The ANALYZE and REPRODUCE phases prioritize *Diagnostics* (∼60%) to pinpoint root causes from failure patterns. (2) Implementation Shift: The EDIT phase transitions to *Implementation* (62%), demanding precise coding patterns for localized fixes. (3) Strategic Verification: VERIFY is driven by *Strategy* (64%), as validation demands high-level meta-reasoning to evaluate test sufficiency and regression risks, distinguishing the logic of quality assurance from the syntax-heavy nature of implementation. This phase-conditioned distribution confirms our core claim: the utility of experience is inherently stage-dependent, allowing our method to route specialized knowledge to the appropriate stage while minimizing cross-phase interference.

### 4.5. Case Study

To explicitly investigate how the proposed method mitigates the granularity mismatch, we conduct a detailed case study on a representative instance from SWE-bench Verified (`sympy__sympy-13757`). This issue concerns reflected multiplication in SymPy, where expressions with a polynomial on the *right-hand side* (e.g., $x * Poly(x)$) do not trigger the intended polynomial multiplication, despite the inverse order behaving correctly. The vanilla agent without memory fails to resolve the issue, repeatedly exploring

the generic multiplication and dispatch logic without pinpointing the reflected-operator pathway. The instance-level memory baseline, which retrieves experiences based on global task-description similarity, also fails but for a different reason. It retrieves broadly related memories (e.g., on parameter-handling control flow and generic edge-case testing) that are semantically similar to the surface text yet functionally irrelevant to the operator-priority and reflected-dispatch mechanism required here.

In contrast, our method aligns retrieval with the agent's functional decomposition. For the ANALYZE subtask, guided by the objective of explaining why the left-side expression fails, the mechanism retrieves a highly specific experience, *Missing Reflected Operator Method*, noting that "when $a * b$ works but $b * a$ fails, check for missing reflected operator methods". For the EDIT subtask, it retrieves a complementary experience, *Python's Dual Operator Implementation Pattern*, recommending implementing the reflected pathway by delegating to the forward implementation to maintain semantic consistency. Leveraging these structure-aligned insights, the agent correctly repairs the reflected multiplication by adhering to SymPy's priority protocol (e.g., `_op_priority` and `call_highest_priority`) and delegating the reflected operation to the existing multiplication routine, successfully resolving the issue, demonstrating the effectiveness of fine-grained retrieval.

### 4.6. Generalization and Reliability Analyses

**Transfer to feature implementation.** The proposed method is intended to transfer as a structural-alignment principle rather than as one fixed taxonomy. To evaluate this, we additionally test on FEA-Bench Lite (Li et al., 2025), which shifts the task family from repository-level bug fixing to repository-level feature implementation. We keep the same agent scaffold, memory mechanism, and inference

budget, and adapt only the prompt and functional taxonomy from ANALYZE / REPRODUCE / EDIT / VERIFY to ANALYZE / DESIGN / IMPLEMENT / VALIDATE. On the 183-instance runnable subset out of 200 Lite instances, Gemini 2.5 Flash improves from 28 to 34 solved instances on average across three runs (15.3% to 18.6%, +3.3 pp). This result supports that what transfers is not a universal four-stage workflow, but the alignment of memory storage, retrieval, and update with task-relevant functional subtasks.

**Robustness to non-canonical workflows.** The four categories define dynamic functional modes for memory routing, not a rigid execution script. Across 11 complete traced runs on SWE-bench Verified, only 53.1% of instances strictly follow the canonical ANALYZE → REPRODUCE → EDIT → VERIFY order. The remaining 46.9% exhibit non-canonical transitions, most commonly skipping RE-PRODUCE when unnecessary (29.2%) or entering iterative EDIT↔VERIFY loops (6.0%). On this non-strict subset, the memory-augmented agent still improves over the matched vanilla baseline (50.0% vs. 46.7%, +3.3 pp), indicating that the benefit does not depend on exact adherence to one fixed stage sequence. A taxonomy-sensitivity analysis further supports this interpretation: on Gemini 2.5 Pro, both a coarser 3-stage variant and a finer 5-stage variant improve over vanilla (57.8% and 57.2% vs. 53.5%), while the proposed 4-stage design performs best (60.3%), suggesting that the benefit is not tied to one unique label set but that granularity still matters.

**Memory quality and online reliability.** We further audit the quality of accumulated and retrieved memories. In a static audit of 200 sampled memory entries, category alignment, transferability, and actionability all remain above 95%, with no evidence of temporal deterioration. In a dynamic audit of 187 retrieved memories, 63.6% are directly relevant, 19.8% are partially relevant, 16.6% are irrelevant, and only 4.8% are judged likely misleading. We also evaluate an offline fixed-memory variant on a repository-disjoint split of SWE-bench Verified, where memories are built from 296 instances and retrieval-only evaluation is performed on the remaining 204 instances. On Claude 3.7 Sonnet, this fixed-memory variant improves from 110 to 118 solved instances, suggesting that the gain is attributable to the memory mechanism itself rather than only to online accumulation.

## 5. Limitations

This work has several limitations. First, although we instantiate subtask-level memory with ANALYZE / REPRO-DUCE / EDIT / VERIFY for repository-level bug fixing, this taxonomy should be viewed as a task-specific functional decomposition rather than a universal workflow ontology. Other software engineering tasks or broader agent domains may require different category spaces, as long as memory storage, retrieval, and update remain aligned with the agent's functional subtasks. Second, our evaluation is still centered on repository-level software engineering tasks with executable correctness signals, and broader validation across more agent frameworks, task families, and long-horizon environments remains an important direction. Third, the method relies on an LLM-based extractor to abstract reusable experience from subtask trajectories. While our analyses suggest that most extracted memories are aligned, transferable, and actionable, noisy memories can still arise from overspecific, overly generic, causally incorrect, or phase-misaligned abstractions. Finally, the current implementation uses an append-only memory state to isolate the effect of structural alignment, but this does not fully address lifelong memory maintenance. Future work should study quality-aware filtering, consolidation of near-duplicate entries, forgetting mechanisms for rarely useful memories, and scalable indexing as the memory bank grows.

## 6. Conclusion

In this paper, we propose a structurally aligned subtask-level memory method to address the fundamental granularity mismatch in existing software engineering agents. By structurally aligning memory retrieval with the agent's functional decomposition, our method decouples actionable reasoning experience from global task descriptions, mitigating reasoning interference in compositional workflows. Extensive experiments on SWE-bench Verified demonstrate that our method delivers consistent performance gains across diverse backbone models. As interaction steps increase, the performance improvement becomes more evident, highlighting the importance of incorporating past experience for effective long-horizon reasoning in complex software engineering tasks. Additional analyses further suggest that these gains do not depend on one rigid stage sequence or fixed taxonomy.

## Impact Statement

Our research focuses on developing memory mechanisms for autonomous software engineering agents using publicly available repository-level benchmarks derived from open-source projects, including SWE-bench Verified and FEA-Bench Lite. We have not performed experiments on human subjects or collected private sensitive data. We rely exclusively on datasets derived from public open-source repositories (GitHub) and ensure compliance with their licenses. Our methods aim to advance the efficiency of automated software maintenance and do not pose foreseeable risks of misuse or societal harm beyond those already inherent to large language models and code generation systems.

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
