# OpenReview forum: "Structurally Aligned Subtask-Level Memory for Software Engineering Agents"
_ICML.cc/2026/Conference — ICML 2026 regular_

### Official Review · Reviewer_Xox9 · 2026-03-05

**Soundness:** 3
**Presentation:** 3
**Significance:** 2
**Originality:** 2
**Overall Recommendation:** 3
**Confidence:** 2

**Summary:**

This paper proposes a subtask-level memory mechanism for software engineering agents. The key argument is that existing memory methods for SWE agents operate at instance level, storing and retrieving entire problem-solving episodes based on global task similarity. The authors argue this causes a granularity mismatch because two tasks that look similar on the surface may require completely different reasoning at specific stages. To fix this, the paper decomposes agent trajectories into subtasks labeled by functional categories like analyze, reproduce, edit, verify, and stores memory entries as triples of category, description, and abstracted experience. During retrieval, the category is used as a hard filter first, then semantic similarity is used to match within that category. During update, an LLM-based extractor abstracts the raw trajectory into transferable insights. Experiments on SWE-bench verified with four backbone models show +4.7 pp improvement on average over vanilla agents.

**Compliance With Llm Reviewing Policy:**

Affirmed.

**Key Questions For Authors:**

See weaknesses

**Limitations:**

yes

**Strengths And Weaknesses:**

Strengths:
(1) Soundness: the experiments are conducted on a well-reputed benchmark.
(2) Significance: the observation that instance-level memory can actually hurt performance on some models is interesting.
(3) Soundness: the analysis in section 4.4 is reasonable. The temporal dynamics plot in figure 3 showing a cold start phase followed by accelerated gains as memory accumulates makes intuitive sense. The complexity-stratified analysis in figure 4 showing the largest gains on hard tasks is also a useful finding.

Weakness:
Originality: the method does not feel very principled to me. The choice of four categories, analyze, reproduce, edit, verify, seems ad hoc and specific to the SWE-bench setting. Why these four? What if the agent's workflow does not follow this structure? The paper does not discuss how to adapt the categories to other types of agent tasks.

---

> ### Author Rebuttal · Authors · 2026-03-31
>
> Thank you for the thoughtful review. We appreciate your recognition of the temporal analyses and the nuanced finding that instance-level memory can sometimes hurt performance. We believe your concern points to one distinction that the original draft did not make explicit enough.
>
> **Our core claim is structural alignment, not a fixed four-stage ontology.** The paper’s core contribution is not a fixed ANALYZE / REPRODUCE / EDIT / VERIFY ontology, nor a method specific to repository-level bug fixing. Rather, the contribution is a structurally aligned memory principle: memory should be stored, retrieved, and updated at the level of task-relevant functional subtasks, instead of whole-instance similarity. This is also reflected in our method formulation, where the memory unit is the structured triplet $(z,d,e)$ of functional category, localized intent, and abstracted experience, rather than a task-specific global episode. In the current paper, we instantiate this principle in repository-level bug fixing because it provides a controlled long-horizon setting with clear task structure and executable correctness signals, making it a suitable testbed for isolating the effect of memory design. More generally, however, the mechanism is modular while the taxonomy is task-adapted; e.g., a long-horizon research agent might naturally use PLAN / RETRIEVE EVIDENCE / SYNTHESIZE / VERIFY. Thus, the intended claim is not that these four labels are universal, but that memory becomes more reusable when its storage and retrieval structure is aligned with the functional subproblems the agent is actually solving.
>
> **Why these four, rather than some other label set?** The choice is not ad hoc, but a granularity trade-off for this task family. It is also grounded in the workflow prior already present in the Mini SWE Agent prompt, which is if anything slightly more fine-grained. To test whether the gain depends on this exact label set, we evaluated both a coarser and a finer taxonomy on SWE-bench Verified: (i) a 3-stage variant that merges REPRODUCE into ANALYZE, yielding ANALYZE / EDIT / VERIFY; and (ii) a 5-stage variant that separates EDGE_CASE from VERIFY. On Gemini 2.5 Pro, both alternatives still improve over the vanilla agent (3-stage: 57.8, 5-stage: 57.2, vs. vanilla: 53.5), while the proposed 4-stage version performs best (60.3). This suggests that the benefit is not tied to one uniquely hard-coded label set, while also showing that taxonomy granularity matters. Our analysis suggests that the 3-stage variant increases within-category interference by mixing diagnostic reasoning with failure-triggering procedures, whereas the 5-stage variant creates sparser and partially overlapping memory pools. For SWE-bench-style bug fixing, the 4-stage taxonomy provides the best trade-off between functional specialization and memory density.
>
> **What if the agent’s realized workflow does not follow the canonical four-stage order?** We also want to clarify that the proposed 4-stage design is not a rigid scripted workflow, but a set of dynamic functional modes under autonomous transition prediction. We analyzed 11 runs with complete phase-transition traces under the same 4-phase memory setting (5,500 traced instances in total). Only 53.1% of instances strictly follow the canonical A→R→E→V sequence, while 46.9% do not. Common deviations include skipping REPRODUCE (29.2%) and iterative EDIT↔VERIFY loops (6.0%). On this non-strict subset, the memory-augmented agent still achieves a 50.0% resolve rate, compared with 46.7% for the matched baseline subset (+3.3 pp). This shows that the four phases should be understood as adaptive functional memory partitions rather than a hard-coded execution template.
>
> **How does this adapt beyond bug fixing?** Following the reviewer’s suggestion, we additionally evaluated on FEA-Bench Lite, a benchmark for repository-level feature implementation derived from real GitHub pull requests spanning 48 repositories. This expands the empirical scope in the two directions raised by the reviewer: it introduces a second benchmark and tests transfer to a different repository-level SWE task family beyond bug fixing. To keep the comparison controlled, we kept the same agent scaffold, memory mechanism, and inference budget, and adapted only the prompts and functional taxonomy to ANALYZE / DESIGN / IMPLEMENT / VALIDATE. On the 183-instance runnable subset, Gemini 2.5 Flash improves from 28 to 34 solved instances on average across three runs (15.3% → 18.6%, +3.3 pp). This supports our intended claim: what transfers is the structurally aligned memory principle, while the taxonomy itself is task-adapted rather than universal.
>
> We will revise the paper to make this distinction much clearer: the central contribution is task-aligned functional memory decomposition, and the current 4-stage design is one validated instantiation for repository-level bug fixing rather than a universal ontology.

---

> > ### Author Rebuttal · Reviewer_Xox9 · 2026-03-31
> >
> > The rebuttal fully resolved my questions

---

> > > ### Author Response · Authors · 2026-04-05
> > >
> > > Thank you again for the thoughtful review and for taking the time to read our rebuttal so carefully. We are especially grateful that you found the concerns fully resolved.
> > >
> > > Your questions helped us clarify the paper’s central claim and improve how we frame its scope and generality. We also appreciate the opportunity to make the discussion of taxonomy adaptation and transfer beyond the current benchmark setting much more explicit.
> > >
> > > We will make these clarifications and the additional supporting analyses explicit in the final version. If, in your judgment, these clarifications and added evidence strengthen the paper’s overall case, we would of course be grateful if that could be reflected in your final assessment. In any case, we sincerely appreciate your careful engagement and helpful feedback.

---

### Official Review · Reviewer_EP1w · 2026-03-12

**Soundness:** 3
**Presentation:** 3
**Significance:** 3
**Originality:** 3
**Overall Recommendation:** 4
**Confidence:** 4

**Summary:**

Existing memory systems for coding agents typically construct and retrieve memories based on global task similarity, which fails when tasks share surface descriptions but require different reasoning logic. The authors decompose the instance-level task into discrete subtasks and store memory entries as triples containing a functional category, localized intent, and experience. Retrieval employs a two-stage strategy: filtering by category as a hard constraint, followed by semantic matching for the specific intent. Evaluations on SWE-bench show that the subtask-level memory outperforms vanilla agents and instance-level baselines, particularly on complex tasks.

**Compliance With Llm Reviewing Policy:**

Affirmed.

**Final Justification:**

I maintain my assessment of weak accept

**Key Questions For Authors:**

1. Figure 5 illustrates the memory retrieval frequency for already retrieved memories. But in general how often are all constructed memories actually used? It would be good to have some discussions on the trade off between memory construction for coverage versus the storage costs for memories that are rarely or never used.
2. Given the long-tail retrieval frequency distribution observed, have you considered implementing a memory consolidation or forgetting mechanism to prune unused memories?

**Limitations:**

No, see weaknesses and key questions above for limitations.

**Strengths And Weaknesses:**

**Strength**:

1. The paper has spotted a clear bottleneck on the granularity of the memories constructed for coding agents. It demonstrated with empirical results that decomposing the memory into sub-task memories help with performance.
2. The paper includes a good range of ablation studies and analysis, such as the impact of category isolation and distribution of memory types, that clearly demonstrate how various design choices contribute to the model's success, and helps to understand what types of memories are constructed and how they align with the problem-solving lifecycle across different phases.

**Weaknesses**

1. The scope of the paper is somewhat narrow, as it is strictly limited to procedural memories for coding agents.
2. The paper only evaluated their method on one dataset. Given the limited scope, it would strengthen the paper's claims to demonstrate efficacy on at least one more dataset, ideally one featuring longer execution trajectories.
3. The paper does not provide an analysis of the efficiency of memory construction and inference. Specifically, it would benefit from a discussion on the computational overhead of extracting and storing memories and the resulting impact on inference latency.

---

> ### Author Rebuttal · Authors · 2026-03-31
>
> Thank you for the careful and constructive review. We appreciate your recognition of the memory-granularity bottleneck identified in the paper and of the accompanying ablations and analyses. We respond to your concerns below.
>
> **(1) Evaluation scope beyond the current benchmark [W1&W2]**
>
> We agree that the current empirical scope is still relatively narrow. The paper’s core contribution is not a fixed ANALYZE / REPRODUCE / EDIT / VERIFY ontology, nor a method specific to coding agents, but a structurally aligned memory principle: memory should be stored, retrieved, and updated at the level of task-relevant functional subtasks rather than whole-instance similarity. This is reflected in the memory unit $(z,d,e)$, which stores functional category, localized intent, and abstracted experience rather than a task-specific global episode. We instantiate this principle in repository-level bug fixing as a controlled testbed with well-defined task structure and executable correctness signals. More generally, the mechanism is modular and the taxonomy task-adapted; e.g., a long-horizon research agent might instead use PLAN / RETRIEVE EVIDENCE / SYNTHESIZE / VERIFY. Thus, the design principle is not tied to one fixed workflow or to coding agents alone.
>
> To broaden the evaluation, we additionally tested on FEA-Bench Lite, which shifts the task family from repository-level bug fixing to repository-level feature implementation. We kept the same agent scaffold, memory mechanism, and inference budget, and adapted only the prompts and taxonomy to ANALYZE / DESIGN / IMPLEMENT / VALIDATE. On the 183-instance runnable subset, Gemini 2.5 Flash improves from 28 to 34 solved instances on average across three runs (15.3% → 18.6%, +3.3 pp). This therefore does more than add a second benchmark: it shows that the proposed memory design transfers to a different repository-level SWE task when instantiated with a task-adapted subtask decomposition.
>
> **(2) Computational overhead of memory construction and retrieval [W3]**
>
> We agree that the memory-specific overhead should be discussed more explicitly. On the retrieval side, the overhead is lightweight by design: memory embeddings are precomputed and kept in memory, retrieval first applies category filtering, and similarity is computed only within the resulting subset. In the current experiments, the candidate set after filtering is still small, so retrieval adds only a small amount of computation relative to a full multi-step agent rollout.
>
> On the construction side, memory writing is triggered only after a subtask is completed, not at every interaction step. It consists of one extractor call, one embedding computation, and appending the resulting memory item. In our implementation, this process is handled concurrently rather than on the agent’s critical reasoning path, so it does not directly delay the next action. Finally, extractor calls are already counted within the same task-level budget limit used for evaluation, so the reported gains are achieved under the same overall budget rather than through extra uncounted computation.
>
> **(3) Memory usage distribution, storage trade-off, and future consolidation [Q1 & Q2]**
>
> We agree that Figure 5 should be interpreted carefully. It reports retrieval frequency conditioned on memories that were retrieved at least once, rather than utilization of the full memory state. In the same representative run, out of 1,731 constructed memories, about 54% are never retrieved, 23% are retrieved once, and 23% are retrieved multiple times, showing a clear head-tail structure beyond the retrieved subset in Figure 5.
>
> We do not interpret low-frequency usage as evidence of low quality. In a static LLM-as-a-judge audit of 200 sampled memory entries, category alignment, transferability, and actionability all remain above 95%, with no evidence of temporal degradation. In a dynamic audit of 187 retrieved memories, retrieved-once entries are not less useful than repeatedly retrieved ones; they receive higher relevance judgments and lower misleading rates. Overall, 63.6% of retrieved memories are directly relevant, 19.8% are partially relevant, and only 4.8% are likely misleading. Thus, the long tail is better understood as a mixture of broadly reusable memories and high-precision specialized memories, rather than low-quality entries, consistent with the paper’s cold-start temporal analysis.
>
> In the current setting, this trade-off is acceptable because each memory is stored as a compact abstracted triple rather than a raw trajectory or long context. We intentionally use a simple append-only state to isolate the effect of subtask-aligned storage and retrieval without adding pruning or consolidation heuristics. We nonetheless agree that explicit maintenance is an important next step, especially for larger or longer-running deployments; natural directions include filtering persistently low-quality entries and consolidating semantically redundant memories.

---

> > ### Author Rebuttal · Reviewer_EP1w · 2026-04-02
> >
> > Thank you for adding the results on FEA-bench and the analysis of memory usage distribution. However, I am not convinced that the computational overhead of memory extraction and storage and its impact on inference latency is trivial. While this overhead may appear negligible on the datasets evaluated here, which are relatively small, it is unclear how it would scale with larger data. In particular, the organisation of memory (and the lack of consolidation or trimming) could significantly affect the storage–usage trade-off as well as the quality and efficiency of retrieval. I understand that addressing these concerns may require more extensive experimentation than can reasonably be included in a rebuttal. Nevertheless, based on the current scope of the paper, I maintain my assessment of weak accept.

---

> > > ### Author Response · Authors · 2026-04-07
> > >
> > > Thank you again for the careful follow-up. We are grateful that you found the added FEA-Bench results and memory-usage analysis helpful, and we also appreciate your further concern regarding memory cost, especially at larger scales. We therefore conducted additional analyses to quantify these costs directly.
> > >
> > > **Extractor overhead.** Extraction is relatively sparse: across the four backbones, memory writing is triggered only 3.2–4.0 times per instance on average (Gemini 2.5 Flash: 3.20, Gemini 2.5 Pro: 3.75, Claude 3.7 Sonnet: 3.49, Claude 4.0 Sonnet: 4.00). In the original pipeline, extractor inputs average 1.6k–2.8k words per subtask trajectory. After filtering verbose bash outputs while retaining the user input, the agent’s responses, and the executed commands, this drops to 0.8k–1.5k words, a 44%–53% reduction. This compression does not materially weaken the memory signal: static memory quality remains above 95% on all three dimensions, dynamic retrieval quality remains similar (61.8% directly relevant vs. 63.6% originally; 5.0% misleading vs. 4.8%), and full SWE-bench Verified performance across all four backbones remains on par, with no material degradation in Pass@1. This suggests that a substantial fraction of extractor burden comes from verbose execution traces rather than from the subtask-memory formulation itself, and can be substantially reduced through task-specific filtering. Similar task-specific filtering may also be possible in other long-horizon settings with structured subtask workflows.
> > >
> > > **Storage overhead.** Each stored entry is a compact structured item rather than a raw trajectory. Across the four backbones, the description field contains only 157–191 characters on average, and the abstracted experience contains 370–479 characters, so the total stored textual content per entry is only 527–661 characters. Relative to the original subtask trajectory, this corresponds to a 95.7%–97.7% reduction in stored textual length. Each entry also includes a fixed 768-dimensional embedding vector. Thus, in the current setting, storage overhead is driven more by the growth in the number of accumulated entries than by the size of each individual stored item.
> > >
> > > **Impact on inference latency.** In our implementation, memory writing is not on the agent’s critical reasoning path: once a subtask boundary is detected, extraction and storage are launched asynchronously in a separate thread, while the main agent continues the SWE interaction. Thus, these operations do not directly block the next action. At the same time, asynchrony does not eliminate the underlying compute cost; it only separates critical-path delay from background compute overhead. Importantly, extractor calls are still counted within the same overall task-level budget used in evaluation, so the reported gains are not obtained through extra uncounted computation.
> > >
> > > **Scaling with larger memory states.** We benchmarked routed-category retrieval latency under increasing memory state sizes. In our benchmarked implementation, initial loading takes 223.5 ms in total, including memory JSONL parsing, embedding JSONL parsing, and NumPy matrix construction. At the current routed-category scale (499 entries), retrieval takes 0.52 ms per query on average, with an embedding-matrix footprint of 1.46 MB. When scaling the routed state by 10× to 5000 entries, latency increases to 5.77 ms per query and the embedding-matrix footprint to 14.65 MB. Thus, the reviewer’s scalability concern is well taken: retrieval cost does grow with state size in the current implementation. At the same time, under the routed-category retrieval setup evaluated here, retrieval remains manageable in the tested range and does not emerge as a dominant bottleneck.
> > >
> > > **Claim boundary and limitation.** These analyses sharpen the boundary of our claim. Our contribution is not an optimized lifelong memory system, but the observation that memory becomes more reusable when storage, retrieval, and updating are aligned with task-relevant functional subtasks rather than whole-instance similarity. Accordingly, we intentionally use a simple append-only memory state with category-routed retrieval to isolate the effect of granularity alignment itself, rather than conflate it with pruning, consolidation, or indexing heuristics. We also agree that a fuller analysis of efficiency and scalability is important for a more complete assessment of the method and its impact. We acknowledge that the current work has an important limitation: the storage-usage trade-off of an append-only memory state, and the role of consolidation, trimming, and more structured organization at larger scales, remain open issues. In the revised version, we will state this limitation explicitly and expand the discussion of next steps, including quality-based pruning, semantic consolidation of near-duplicate memories, and finer-grained organization within each functional category to improve retrieval efficiency and quality.

---

### Official Review · Reviewer_59Fm · 2026-03-13

**Soundness:** 3
**Presentation:** 3
**Significance:** 3
**Originality:** 3
**Overall Recommendation:** 4
**Confidence:** 4

**Summary:**

This paper proposes a subtask-level memory mechanism for SWE agents. Instead of storing and retrieving entire problem-solving episodes (instance-level memory), the method decomposes the agent's workflow into functional subtasks (Analyze, Reproduce, Edit, Verify), stores abstracted insights per subtask category, and retrieves them with category isolation. Evaluated on SWE-bench Verified with multiple backbones, the method improves Pass@1 by +4.7pp on average over the vanilla agent, with the largest gains on hard tasks (+8.7%).

**Compliance With Llm Reviewing Policy:**

Affirmed.

**Final Justification:**

My concerns have been solved.

**Key Questions For Authors:**

1. The 4-stage decomposition (Analyze/Reproduce/Edit/Verify) is tailored to bug-fixing. Have you tested on any other task format? If not, what evidence do you have that the method transfers?

2. What fraction of extracted insights are actually useful vs. noisy? The paper shows 50.7% of memories are single-use (Figure 5) Does this mean they helped, or that they were retrieved but irrelevant? Can you distinguish between "retrieved and helped" vs. "retrieved and ignored"?

3. Can you provide p-values or confidence intervals for the main results in Table 1? The improvement is consistent across models but some individual gains are small relative to variance.

4. How sensitive is the method to the order in which the 500 instances are processed? Have you run with different random orderings? The cold-start analysis (Figure 3) suggests ordering matters. Could an adversarial ordering degrade performance?

**Limitations:**

The paper discusses limitations minimally, only in the Impact Statement, which focuses on data ethics rather than methodological limitations. The fixed decomposition assumption, single-benchmark evaluation, and ordering sensitivity are not discussed.

**Strengths And Weaknesses:**

**Strengths:**

- S1: The "granularity mismatch" argument is well-articulated. The paper demonstrates concretely (Section 4.5, sympy-13757 case study) how instance-level retrieval finds semantically similar but functionally irrelevant memories, while subtask-aligned retrieval finds the right insight at the right stage.

- S2: The ablation studies (Tables 2-4) cleanly isolate contributions: structured prompting alone gives only +1.0pp, global retrieval gives +1.6pp, raw trajectories give +1.2pp, while the full method gives +3.9pp. This makes the contribution claims credible.

**Weaknesses:**

- W1: The method hardcodes the workflow into Analyze, Reproduce, Edit, Verify. This may not generalize well to tasks outside SWE-bench's issue-resolution format, e.g., feature implementation, code review, or refactoring. The paper does not discuss this limitation at all. Real SWE tasks don't always follow this decomposition (see FEA-bench, cited in the paper).

- W2: The "extraction operator" that converts raw trajectories into abstract insights is described only at a high level. It appears to be an LLM call that summarizes what was learned. How sensitive is the method to the quality of this abstraction? What does a bad abstraction look like, and how often does it happen? The paper doesn't analyze failure cases of the abstraction step.

- W3: The entire evaluation is on SWE-bench Verified (500 instances). While this is a standard benchmark, evaluating on a single benchmark limits the generalizability claims. Concurrent work like Multi-SWE-bench or FEA-bench would strengthen the evaluation.

- W4: The method operates in an online, sequential fashion, memory accumulates as the agent processes tasks. But the order matters (acknowledged implicitly by the cold-start analysis). The paper doesn't compare against an offline variant where all memories are pre-populated, which would help disentangle "memory helps" from "online learning helps".

- W5: Table 1 shows standard deviations, but some are large. With 500 test instances, a 3.9pp improvement with delta in the 1-2 range is not overwhelmingly significant. Statistical significance tests (e.g., paired bootstrap) are not reported.

---

> ### Author Rebuttal · Authors · 2026-03-31
>
> Thank you for the careful and constructive review. We appreciate your recognition of the paper’s core motivation and the strength of the ablation studies. We respond to your concerns below.
>
> **(1) Transfer beyond bug fixing and beyond a single benchmark [W1&W3&Q1]**
>
> We agree that the original empirical scope was limited. Our core claim, however, is not that ANALYZE / REPRODUCE / EDIT / VERIFY is a universally fixed workflow, but that memory should be stored, retrieved, and updated at the level of task-relevant functional subtasks rather than whole-instance similarity. Accordingly, the method only requires a task-appropriate category/description space for the memory triple $(z,d,e)$, not one unique label set.
>
> Following this suggestion, we additionally evaluated on FEA-Bench Lite. To keep the comparison controlled, we used the same agent scaffold, memory mechanism, and inference budget, and adapted only the prompts and functional taxonomy to ANALYZE / DESIGN / IMPLEMENT / VALIDATE. On the matched 183-instance runnable subset (17 Lite instances were excluded for setup reasons unrelated to our method, including missing historical commits for 14 `tobymao/sqlglot` tasks), Gemini 2.5 Flash improves from 28 to 34 solved instances on average across three runs (15.3% → 18.6%, +3.3 pp).
>
> **(2) Extraction quality and whether retrieved memories are actually useful [W2&Q2]**
>
> To address this concern, we conducted two complementary LLM-as-a-judge analyses. First, we statically audited 200 sampled memory entries produced by the extractor. The extracted memories consistently satisfy basic quality constraints: category alignment, transferability, and actionability all remain above 95%, suggesting that the abstraction step is generally reliable at the text level.
>
> Second, we audited the memories actually retrieved for later subtasks along two dimensions: relevance and expected utility. Among 187 retrieved memories, 63.6% are directly relevant, 19.8% are partially relevant, and only 4.8% are judged likely misleading. Thus, retrieval is not perfect, but the dominant failure mode is neutrality rather than harm. This also clarifies Figure 5: “single-use” refers only to retrieval frequency among retrieved memories, not whether a memory helped. In an online stream, frequency is also affected by retrieval opportunity, and in our audit, retrieved-once memories are not less useful than repeated ones, and are often narrower and more precise. Failure cases are mostly interpretable: retrieval errors typically arise from partial-similarity but wrong-mechanism matches, while extraction errors are mainly overspecific, overly generic, causally incorrect, or phase-misaligned abstractions. These errors point to more precise matching and filtering as natural next steps.
>
> **(3) Ordering sensitivity and online vs. offline memory [W4&Q4]**
>
> We agree that processing order matters in a streaming memory setup: the memory state starts empty and is populated online, so earlier tasks affect what later tasks can retrieve. However, the main results are not tied to a single execution order. As described in the paper, every result in Table 1 is averaged over three independent runs with different random shuffles of the 500-instance stream, and the gains remain consistent across backbones and seeds. Thus, order affects the magnitude of the gain, but the improvement is not an artifact of one favorable ordering. We also do not claim full order-invariance: an adversarial ordering could in principle prolong the cold-start phase, but we do not observe collapse under the random orderings tested.
>
> To separate “memory helps” from “online learning helps,” we additionally ran an offline fixed-state variant on a repo-disjoint 296/204 split of SWE-bench Verified. We built a fixed memory state from the 296-instance subset and evaluated the 204-instance subset with retrieval only and no further updates. On Claude 3.7 Sonnet, this offline variant still improves over vanilla (110 → 118), showing that the benefit is attributable to the memory mechanism itself rather than only to online accumulation. At the same time, the original online setting remains important because it provides continual adaptation beyond a fixed state.
>
> **(4) Statistical significance of the main gains [W5&Q3]**
>
> We further evaluated statistical significance using exact McNemar tests on paired per-instance pass/fail outcomes. For each backbone, we formed three run pairs between our method and the vanilla agent and computed significance on the 500 SWE-bench Verified instances within each pair. All 12/12 comparisons satisfy p < 0.05, including Claude 4.0 Sonnet despite its smaller mean gain. These results support that the gains in Table 1 are statistically reliable and unlikely to be explained by random fluctuation alone.
>
>
> We will also make the methodological limitations more explicit in the revision.

---

> > ### Author Rebuttal · Reviewer_59Fm · 2026-04-07
> >
> > My concerns have been solved. Thank you for the rebuttal.

---

> > > ### Author Response · Authors · 2026-04-07
> > >
> > > Thank you again for the careful and constructive review, and for taking the time to read our rebuttal so closely. We are very grateful that you found your concerns fully resolved.
> > >
> > > Your comments helped us clarify the paper’s scope and positioning, and strengthen several parts of the empirical analysis.
> > >
> > > We will incorporate these clarifications and additional analyses into the final version so that the paper more clearly communicates both its current empirical scope and its broader methodological takeaway. Thank you again for the thoughtful feedback and for helping us improve the paper.

---

### Official Review · Reviewer_vm9v · 2026-03-18

**Soundness:** 2
**Presentation:** 3
**Significance:** 2
**Originality:** 3
**Overall Recommendation:** 4
**Confidence:** 4

**Summary:**

This paper studies memory mechanisms for software engineering agents and argues that existing instance-level memory approaches suffer from a challenge: they retrieve experience at the level of whole episodes, whereas software engineering tasks are naturally composed of heterogeneous subtasks such as analysis, reproduction, editing, and verification. To address this mismatch, the paper proposes Structurally Aligned Subtask-Level Memory, which stores and retrieves experience at the subtask level. Each memory item is represented as a triple of subtask category, structured description, and abstracted experience, and retrieval is performed in two stages: category filtering followed by semantic matching. The method also updates memory online by extracting reusable insights from completed subtasks. Experiments on SWE-bench Verified across four backbone models show consistent improvements over both a vanilla agent and an instance-level memory baseline, with an average gain of +4.7 Pass@1 points over the vanilla agent and up to +6.8 points on Gemini 2.5 Pro. The paper further shows that gains are larger on harder, longer-horizon tasks.

**Compliance With Llm Reviewing Policy:**

Affirmed.

**Key Questions For Authors:**

How robust is the method to segmentation mistakes, especially when the agent predicts the wrong subtask transition or mislabels the current intent?
Does the quality of the extracted memory content degrade over long streams, or does the memory state remain stable as more experiences accumulate?
Can the authors comment on how well this method would transfer to other agent benchmarks beyond SWE-bench Verified?

**Limitations:**

yes

**Strengths And Weaknesses:**

Strengths:
This paper addresses an important problem for long-horizon software engineering agents: how to design memory at the right granularity for effective experience reuse. A central context analyzed by this study is the mismatch between monolithic, instance-level retrieval and the stage-structured nature of repository-level issue resolution. This motivation is clearly illustrated in Figure 1 on page 1, which contrasts misleading global similarity with more precise subtask-aligned retrieval.
The proposed method is intuitive and well structured. The decomposition into four subtask categories—ANALYZE, REPRODUCE, EDIT, and VERIFY—combined with category-constrained retrieval is a simple but well-motivated design choice. Figure 2 on page 3 clearly explains the retrieval/update loop, and the algorithmic description is easy to follow.
The empirical results are strong and consistent. In Table 1 on page 5, the method improves over both the vanilla agent and the instance-level memory baseline across all four backbones, including Gemini 2.5 Flash, Gemini 2.5 Pro, Claude 3.7 Sonnet, and Claude 4.0 Sonnet. The ablations in Tables 2–4 on page 6 show that gains do not come merely from structured prompting, but from the combination of subtask-level retrieval, category isolation, and abstracted memory content.

Weaknesses:
My main concern is the scope of evaluation. The experiments are conducted only on SWE-bench Verified, using a single agent (Mini SWE Agent). While this is a strong and relevant benchmark, it is still unclear how much the conclusions generalize to other software engineering settings  or broader agent environments. The manuscript would be stronger with at least one additional benchmark or a more extensive cross-framework validation.
A second concern is that the subtask decomposition is manually restricted to four categories. This seems reasonable and practical, but it is not fully clear whether these categories are universally appropriate across repositories and issue types, or whether the method is sensitive to the chosen taxonomy. Since the agent predicts transitions autonomously, errors in segmentation could affect both retrieval quality and memory updates, but this issue is not deeply quantified.
A third concern is the dependence on the quality of the extracted abstract experience. The paper argues convincingly that abstraction is better than raw trajectories, and the ablation supports this. However, the extraction operator uses the same backbone model as the solving agent, and there is limited direct analysis of extraction errors or noisy memory accumulation over long streams. It would be useful to know whether the memory base degrades, stabilizes, or self-corrects over time.

---

> ### Author Rebuttal · Authors · 2026-03-31
>
> Thank you for the careful and constructive review. We appreciate your recognition of the paper’s core motivation, design clarity, and consistent gains across models and ablations. We respond to your concerns below.
>
> **(1) Evaluation scope and transfer beyond SWE-bench Verified**
>
> We agree that the original empirical scope was limited, and we will state this boundary more explicitly in the revision. Our core claim is not that ANALYZE / REPRODUCE / EDIT / VERIFY is a universally fixed workflow, but that memory should be stored, retrieved, and updated at the level of task-relevant functional subtasks rather than whole-instance similarity. Accordingly, the method only requires a task-appropriate category/description space for the memory triple $(z,d,e)$, not one unique label set. We instantiate this principle on repository-level bug fixing because it is a well-established long-horizon SWE task and SWE-bench Verified provides rigorous executable correctness signals.
>
> Following this suggestion, we additionally evaluated on FEA-Bench Lite, which shifts the task family from bug fixing to repository-level feature implementation. We kept the same agent scaffold, memory mechanism, and inference budget, and adapted only the prompts and functional taxonomy to ANALYZE / DESIGN / IMPLEMENT / VALIDATE. On the 183-instance runnable subset, Gemini 2.5 Flash improves from 28 to 34 solved instances on average across three runs (15.3% → 18.6%, +3.3 pp). This both adds a second benchmark and demonstrates transfer across SWE task families when the taxonomy is adapted accordingly. We will revise the paper to clarify this point: what transfers is the structural-alignment principle, while the taxonomy itself is task-adapted rather than universal.
>
> **(2) Generality and robustness of the four-stage decomposition**
>
> The four categories are intended as a task-specific functional decomposition for repository-level bug fixing, not a universal or rigid workflow template. What matters is that retrieval and update are aligned with the agent’s current functional mode rather than whole-instance similarity.
>
> This is also reflected in realized trajectories. In an auxiliary analysis of 11 complete runs (5,500 traced instances), only 53.1% of instances strictly follow the canonical A→R→E→V order, while 46.9% do not, most commonly by skipping REPRODUCE when unnecessary (29.2%) or entering iterative EDIT↔VERIFY loops (6.0%). On this non-strict subset, the memory-augmented agent still outperforms the matched baseline (50.0% vs. 46.7%, +3.3 pp), showing that the benefit does not depend on exact adherence to one fixed stage sequence. This is consistent with our additional taxonomy-sensitivity analysis (discussed in more detail in R4): coarser and finer variants still improve over vanilla, while the proposed 4-stage design performs best.
>
> For segmentation errors, our case analysis suggests that fully wrong category assignments are relatively uncommon. The more typical issue is either boundary ambiguity, with slightly early or late transitions, or EDIT↔VERIFY oscillation, which is more consequential and more common on Gemini models. To limit propagation from such unstable transitions, the experiments use a simple implementation-level safeguard: within a task, repeated memory operations for the same category are capped, so repeated EDIT↔VERIFY switching does not keep re-triggering redundant retrieval or updates. We will make this implementation detail explicit in the revision.
>
> **(3) Stability of extracted memory over long streams**
>
> We agree with the reviewer’s concern that the method depends on extractor quality and that a key question is whether noisy memories accumulate over long streams. We therefore conducted two complementary LLM-as-a-judge analyses. First, we statically audited 200 sampled memory entries produced by the extractor. Across early, mid, and late periods, the extracted memories consistently satisfy basic quality constraints: category alignment, transferability, and actionability all remain above 95%, with no evidence of temporal deterioration. This suggests that newly written memories remain stable in textual quality rather than progressively degrading as the stream grows.
>
> Second, we audited the memories actually retrieved for later subtasks. Overall, retrieval quality is high: among 187 retrieved memories, 63.6% are directly relevant, 19.8% are partially relevant, and only 16.6% are irrelevant; only 4.8% are judged likely misleading. This quality also does not worsen over time. Direct relevance rises from 57.1% in the early period to 75.8% in the late period, while likely misleading retrieval decreases from 4.8% to 1.6%. Taken together, these results do not support progressive noise accumulation; instead, they suggest that the memory state remains stable as the bank grows. An important future direction is to add explicit memory maintenance, such as filtering low-quality entries or consolidating near-duplicates.

---

> > ### Author Rebuttal · Reviewer_vm9v · 2026-04-03
> >
> > My concerns are fully resolved.

---

> > > ### Author Response · Authors · 2026-04-05
> > >
> > > Thank you again for the careful and constructive review, and for taking the time to read our rebuttal. We are very grateful that you found your concerns fully resolved.
> > >
> > > Your comments helped us clarify the paper’s scope, strengthen the discussion of robustness to non-canonical subtask transitions, and make the analysis of long-stream memory stability much more explicit. We also appreciate the opportunity to better articulate what is intended to transfer across settings: the structural-alignment principle, while the taxonomy itself is task-adapted.
> > >
> > > We will incorporate these clarifications and additional analyses into the final version so that the paper more clearly communicates both its current empirical scope and its broader methodological takeaway. We sincerely appreciate your thoughtful feedback and any further consideration.

---

### Decision · Program_Chairs · 2026-04-30

**Decision:**

Accept (regular)

**Comment:**

Three reviewers recommend weak accept (4) and one initially recommended weak reject (3). The weak rejecting reviewer (Xox9) explicitly marked their concerns as fully resolved after the rebuttal. The overall recommendation is weak accept.

The paper addresses a concrete limitation of instance-level memory in SWE agents: when tasks share surface-level similarity but require different reasoning at specific stages, whole-episode retrieval misleads the agent. The proposed solution is to align memory storage and retrieval with the agent's functional decomposition (ANALYZE/REPRODUCE/EDIT/VERIFY), storing compact triples of category, intent, and abstracted experience. The motivation is clear, the ablations are clean, and the empirical results are consistent across four backbone models.

The main concerns across reviewers were: (1) evaluation limited to SWE-bench Verified; (2) the four-category taxonomy seeming ad hoc; (3) memory quality and scalability. All three were addressed in the rebuttal. The authors added evaluation on FEA-Bench Lite with an adapted taxonomy, showing +3.3 pp transfer with a different decomposition (ANALYZE/DESIGN/IMPLEMENT/VALIDATE), which supports the general principle. Statistical significance was confirmed via McNemar tests (p < 0.05 across all 12 backbone-run pairs). Memory quality audits showed above 95% quality on stored entries and 63.6% directly relevant retrieval. The offline fixed-memory experiment also confirmed that the gain is from the memory mechanism itself rather than just from online accumulation.

Reviewer EP1w raised a follow-up concern about scalability of the append-only memory state at larger scales — a reasonable concern that the authors acknowledged as a limitation and provided latency benchmarks for. This is a fair limitation to flag in the paper, and the authors committed to expanding the discussion.

Overall, the paper makes a clean and practically useful contribution to SWE agent memory design, with strong empirical grounding and a comprehensive rebuttal. Weak accept.